# Dietary Omega-3 PUFAs in Metabolic Disease Research: A Decade of Omics-Enabled Insights (2014–2024)

**DOI:** 10.3390/nu17111836

**Published:** 2025-05-28

**Authors:** Jing Li, Yang-Chi-Dung Lin, Hua-Li Zuo, Hsi-Yuan Huang, Tao Zhang, Jin-Wei Bai, Hsien-Da Huang

**Affiliations:** 1School of Medicine, The Chinese University of Hong Kong, Shenzhen 518172, China; lijing0903@cuhk.edu.cn (J.L.); yangchidung@cuhk.edu.cn (Y.-C.-D.L.); zuohuali@cuhk.edu.cn (H.-L.Z.); huanghsiyuan@cuhk.edu.cn (H.-Y.H.); 2Warshel Institute for Computational Biology, School of Medicine, The Chinese University of Hong Kong, Shenzhen 518172, China; waynebai319@gmail.com; 3Guangdong Provincial Key Laboratory of Digital Biology and Drug Development, The Chinese University of Hong Kong, Shenzhen 518172, China; 4R&D Center, Better Way (Shanghai) Cosmetics Co., Ltd., Shanghai 201103, China; zhangt@mistinechina.com; 5Department of Endocrinology, Key Laboratory of Endocrinology of National Health Commission, Peking Union Medical College Hospital, Chinese Academy of Medical Sciences & Peking Union Medical College, Beijing 100730, China

**Keywords:** omega-3 polyunsaturated fatty acids, metabolic diseases, omics

## Abstract

**Background/Objectives**: The rising global prevalence of metabolic diseases (e.g., obesity, type 2 diabetes mellitus) underscores the need for effective interventions. Omega-3 polyunsaturated fatty acids (PUFAs) exhibit therapeutic potential, yet their molecular mechanisms remain unclear. This systematic review synthesizes a decade (2014–2024) of omics research to elucidate Omega-3 PUFA mechanisms in metabolic diseases and identify future directions. **Methods**: A PRISMA-guided search of the Web of Science identified studies on Omega-3 PUFAs, metabolic diseases, and omics. After excluding reviews, non-English articles, and irrelevant studies, 72 articles were analyzed (16 multi-omics, 17 lipidomics, 10 transcriptomics/metabolomics/microbiomics each, and 6 proteomics). **Results**: Omics studies demonstrated that Omega-3 PUFAs, particularly EPA and DHA, improve metabolic health through interconnected mechanisms. They regulate epigenetic processes, including DNA methylation and miRNA expression, influencing genes linked to inflammation and insulin sensitivity. Omega-3 PUFAs reduce oxidative stress by mitigating protein carbonylation and enhancing antioxidant defenses. Gut microbiota modulation is evident through increased beneficial taxa (e.g., Bacteroidetes, Akkermansia) and reduced pro-inflammatory species, correlating with improved metabolic parameters. Mitochondrial function is enhanced via upregulated fatty acid oxidation and TCA cycle activity, while anti-inflammatory effects arise from NF-κB pathway suppression and macrophage polarization toward an M2 phenotype. Challenges include interindividual variability in responses and a limited understanding of dynamic metabolic interactions. **Conclusions**: Omega-3 PUFAs target multiple pathways to improve metabolic health. Future research should prioritize chemoproteomics for direct target identification, multi-omics integration, and personalized strategies combining Omega-3 with therapies like calorie restriction.

## 1. Introduction

The escalating burden of non-communicable diseases (NCDs) worldwide has emerged as a significant public health challenge. The annual healthcare cost per patient is projected to exceed USD 5000 in low- and middle-income countries [1,2,3]. This substantial burden is mainly attributable to the increasing prevalence of metabolic diseases. The broad category of “metabolic diseases” includes hypertension, type 2 diabetes mellitus (T2DM), hyperlipidemia, obesity, and non-alcoholic fatty liver disease (NAFLD) [4]. These conditions frequently coexist, share common etiological factors, and collectively increase the risk of disability, cancer development, and premature mortality [5,6]. Utilizing the Global Burden of Disease (GBD) 2021 data [1], we employed the Disability-Adjusted Life Years (DALYs) metric and mortality data to characterize the health burden of five common metabolic diseases: T2DM, hypertension, obesity, hypercholesterolemia, and Metabolic Dysfunction-Associated Steatotic Liver Disease (MASLD) (Figure 1). The results indicate that the burden of these diseases has increased by 1.6 to 3 times over the past three decades. The geospatial analysis reveals that the regions with the highest absolute burden remain concentrated in the world’s most populous countries, particularly India, China, and the USA [1].

Dietary interventions are increasingly gaining attention and have become crucial in preventing and managing metabolic diseases [7,8,9]. Omega-3 polyunsaturated fatty acids (PUFAs) are particularly important due to their numerous health benefits. Substantial evidence indicates that Omega-3 PUFAs, specifically eicosapentaenoic acid (EPA) and docosahexaenoic acid (DHA), exert beneficial effects on multiple metabolic parameters, including triglyceride reduction, insulin sensitivity improvement, and blood pressure regulation [10,11,12]. As essential fatty acids that cannot be endogenously synthesized, the parent Omega-3 fatty acid, ALA, and the parent Omega-6 fatty acid, LA, must be obtained through dietary sources. ALA is primarily found in plant sources such as chia seeds, flaxseeds, and walnuts, while EPA and DHA are predominantly obtained from marine sources, including salmon, mackerel, and sardines (Figure 2a). The limited conversion efficiency of plant-derived ALA to EPA and DHA necessitates direct consumption of marine sources for optimal health benefits [13] (Figure 2b). The global Omega-3 market growth, driven by health demands, has accelerated sustainable alternatives such as algal oil (*Schizochytrium* sp.) [14] and genetically modified plant oils (e.g., EPA/DHA-enriched canola oil) [15,16,17,18].

The distinct biological roles of DHA and EPA in metabolic regulation have been extensively investigated. These marine-derived Omega-3 PUFAs exert their therapeutic effects through multiple mechanisms. Primarily, they are essential in reducing inflammation [19,20], which is critical in obesity, T2DM, and cardiovascular diseases (CVDs). Furthermore, DHA and EPA improve insulin sensitivity [21] and mitigate oxidative stress, which is essential for the prevention and management of T2DM and hypertension [22,23]. These fatty acids can also aid in weight management by promoting fat oxidation and reducing fat storage. Their cardioprotective effects are particularly noteworthy, as they improve lipid profiles and reduce blood pressure, lowering cardiovascular disease risk [24]. However, clinical trials of these agents have yielded mixed results. Based on the available evidence, EPA appears to be more effective in reducing cardiovascular risk compared to the combination of DHA and EPA [25,26,27]. Beyond metabolic and cardiovascular health, DHA and EPA have demonstrated potential in supporting cognitive function and mood regulation, offering additional benefits for individuals with metabolic diseases at risk of cognitive decline or mood disorders [28]. In summary, incorporating DHA and EPA into daily diets or supplements can be considered an effective strategy to combat metabolic diseases, offering a natural and efficient approach to enhance overall health.

ALA is the metabolic precursor for EPA and DHA in the human body. However, this bioconversion process is notably inefficient, with typically less than 8% of dietary ALA converted to EPA and less than 4% to DHA [29]. ALA exhibits many biological functions, including metal ion chelation, reactive oxygen species (ROS) quenching, and regeneration of oxidized glutathione and vitamins C and E [30]. Its enantiomers and reduced form demonstrate diverse therapeutic properties, such as antioxidant, anti-inflammatory, neuroprotective, antimicrobial, and cardioprotective effects, alongside benefits in cancer treatment, fertility enhancement, and metabolic regulation [31]. A systematic review and meta-analysis revealed that ALA supplementation at ≥3 g/d from flaxseed and flaxseed oil significantly improved CVD risk profiles in obese or overweight individuals, particularly with longer intervention (≥12 weeks) and poorer baseline cardiovascular health [32]. Mechanistically, ALA activates the 5′ adenosine monophosphate-activated protein kinase (AMPK) and inhibits nuclear factor kappa B (NF-κB), reducing cholesterol biosynthesis, fatty acid β-oxidation, and vascular stiffness. ALA also promotes insulin secretion, glucose transport, and insulin sensitivity by activating the phosphatidylinositol 3-kinase (PI3K)/Akt pathway. Moreover, ALA addresses central obesity by increasing adiponectin levels and mitochondrial biogenesis, and it can reduce food intake primarily through stimulation of silent information regulator T1 (SIRT1) [33]. These multifaceted mechanisms underscore ALA’s therapeutic potential in managing metabolic diseases and associated comorbidities.

Recent advances in nutrigenomics have revolutionized our understanding of nutrient–gene interactions. The integration of high-throughput omics technologies including transcriptomics, proteomics, metabolomics, and microbiomics with advanced bioinformatics tools has enabled the identification of molecular markers and mechanistic pathways underlying the health benefits of Omega-3 PUFAs [34,35]. This review aims to examine omics approaches in metabolic research from both animal and human studies to understand the effects of Omega-3 PUFAs, analyze the mechanistic relationships between Omega-3 PUFAs and omics, discuss research challenges, and propose future directions.

## 2. Materials and Methods

### 2.1. Data Retrieval

In this study, we utilized the globally recognized comprehensive platform Web of Science (WOS) to compile a dataset of research papers focusing on omics data analysis and Omega-3 PUFAs related to metabolic diseases. We retrieved articles published before 27 November 2024, following the Preferred Reporting Items for Systematic Reviews and Meta-Analyses (PRISMA) guidelines. PRISMA provides an evidence-based minimum checklist for reporting systematic reviews and meta-analyses.

The search query was performed in the WOS core collection. The following search strategy was employed: Topic = (Omega-3 fatty acid* OR *N*-3 Polyunsaturated fatty acid* OR *N*-3 PUFA* OR *N*-3 fatty acid* OR ω-3 fatty acid* OR Omega-3 PUFA* OR ω-3 Polyunsaturated fatty acid*) AND (Metabolic Syndrom* OR Metabolic Disorder* OR Metabolic Condition* OR Metabolic Pathology* OR Metabolic Dysfunct* OR Metabolic Disease* OR Metabolic Abnormalit*) AND (omic* OR Genom* OR Transcriptom* OR Proteom* OR Metabolom* OR Epigenom* Microbiom* OR lipidom* OR profile* OR Sequenc*).

### 2.2. Screening and Exclusion Criteria

Once the publications were retrieved from the WOS database, we compiled a comprehensive list of all the search results. Subsequent systematic integration and screening were conducted according to our previous methods [36,37,38]. Each item was carefully labeled in an Excel spreadsheet with the reason for its exclusion (Table 1).

### 2.3. Annotated Bibliography

After the initial screening phase, all eligible papers were retrieved, and their abstracts or full texts were thoroughly reviewed to assess their relevance. Subsequently, each qualifying research paper was meticulously annotated based on manually extracted data, including research objectives, research methods, animal or human models used, relevant biomolecules (targets or genes), and reported mechanisms of action. Key insights from review articles were also systematically extracted. Finally, statistical analyses were conducted to summarize the current state of research in this field.

## 3. Search Results and Study Characteristics

As illustrated in Figure 3, the literature survey and screening process ultimately led to the inclusion of 72 full-text studies for this review (Figure 3a). Initially, a comprehensive search of the WOS database identified a total of 1311 records. Then, exclusions were carried out according to predefined criteria [36,37]. Specifically, non-research articles (such as reviews and editorials; *n* = 117) were removed, along with articles not published in English (*n* = 26) and publications outside the specified time frame from 2014 to 2024 (*n* = 307). After that, the abstracts of the remaining articles were evaluated for relevance. As a result, studies outside the field of life sciences (*n* = 103) and those with irrelevant subjects or topics (*n* = 693) were excluded. Among the included 72 studies, 16 studies employed multi-omics, 10 used transcriptomics alone, 17 utilized lipidomics, and there were 10 each for metabolomics and microbiomics studies. Genomics and epigenomics together accounted for three papers, with one for genomics and two for epigenomics. Additionally, six studies used proteomics (Figure 3b).

This emerging topic is increasingly garnering widespread attention from the global scientific community. Figure 4 illustrates the geographical distribution of publications focusing on Omega-3 PUFAs in animal or human models based on omic studies, particularly emphasizing their role in the mechanisms of metabolic diseases. The analysis reveals that the USA is the leading contributor to research output. It is followed by China, Spain, Canada, and the Czech Republic, as shown in Figure 4.

## 4. Omics Approaches in Animal Studies

### 4.1. Animal Transcriptomics: Gene Regulatory Mechanisms Mediated by Omega-3 PUFAs

Transcriptomics studies the complete set of RNA transcripts produced by the genome, which can deeply reveal the gene expression and its regulatory mechanisms. Over the past decade, the application of transcriptomics in the study of Omega-3 PUFAs has gained momentum. Key areas include the transgenerational effects of Omega-3 PUFAs, miRNA-mediated regulatory mechanisms, and the integration of transcriptomics with other omics technologies to understand their overall impact on metabolic health holistically [39,40,41,42,43,44,45,46,47,48] (Appendix A).

Varshney et al. [39] utilized mouse models to investigate the effects of early exposure to a low Omega-6/Omega-3 PUFA ratio of postnatal milk consumption from “fat-1” transgenic dams for 10 days. They identified diverse adipocyte precursor cell (APC) subtypes using single-cell RNA sequencing. They demonstrated that early exposure would increase mitochondrial-high APCs content, fatty acid oxidation capacity, and expression of beige adipocyte markers (Ppargc1α, Ucp2, and Runx1) in APCs, which was related to the increase in NR2F2 levels. These findings suggest that early exposure to a low Omega-6/Omega-3 PUFA ratio promotes a thermogenic and obesity-resistant phenotype, highlighting its potential role in shaping long-term metabolic health.

Over the past four decades, the global prevalence of childhood obesity has soared, causing health and social issues and becoming a pressing public health challenge [49]. In a porcine model study [40], researchers examined the effects of a high-calorie western-type diet supplemented with Omega-3 PUFAs over 10 weeks in prepubertal pigs. A transcriptomic analysis revealed that Omega-3 supplementation reduced adipogenesis and inflammation. Conversely, it promoted fatty acid oxidation and cholesterol catabolism. These results indicate that Omega-3 PUFAs can mitigate obesity-related metabolic dysregulation induced by energy-dense diets in early life.

Another study [41] explored the transgenerational effects of Omega-3 PUFA supplementation using a mouse model. Over three generations, offspring were subjected to a 4-week obesogenic diet. A transcriptomic analysis of adipose tissue showed that transgenerational Omega-3 intake reduced the expression of inflammatory genes (Foxo3, Gsk3beta) and mitigated perturbations in metabolic homeostasis, cholesterol metabolism, and mitochondrial function. These findings suggest that long-term Omega-3 intake across generations may protect against obesity-related metabolic disorders, particularly in energy-dense diets.

Omega-3 PUFAs regulate metabolic health through miRNA-mediated mechanisms. In commercial pig diets, excessive Omega-6 PUFAs elevate inflammatory risks, while Omega-3 supplementation counteracts these pro-inflammatory effects [43]. A study altering the Omega-6/Omega-3 ratio demonstrated that Omega-3 modulates key miRNAs (e.g., ssc-miR-30a-3p, ssc-miR-30e-3p, ssc-miR-15b, and ssc-miR-7142-3p), which are linked to adipogenesis, lipolysis, and immune pathways [43]. These miRNAs were part of regulatory networks affecting metabolic pathways such as obesity, myogenesis, and protein degradation, providing insights into how Omega-3 PUFAs modulate metabolic health through miRNA regulation. Karla Fabiola Corral-Jara et al. [48] utilized transcriptomics to study the effects of transgenerational Omega-3 PUFA supplementation, specifically EPA, in mice subjected to an obesogenic diet. They found that EPA mediated sophisticated adaptive alterations at the molecular scale, resulting in the downregulation of miR-34a-5p, a negative regulator of Irs2. Irs2 is a key regulator in hepatic gluconeogenesis, and its proper function is crucial for preventing insulin resistance [48].

### 4.2. Animal Proteomics: Protein Modifications and Pathways Induced by Omega-3 PUFAs

Proteomics, through analyzing the changes in protein levels under dietary intervention, has become an essential tool for understanding how Omega-3 PUFAs affect metabolic health. Recent studies on cow, mouse, and rat models have shown that Omega-3 PUFAs can reduce inflammation [50], improve insulin sensitivity, and ameliorate metabolic diseases by regulating proteins involved in lipid metabolism, carbohydrate metabolism [51], and antioxidant defense [52,53] (Appendix A).

A study used proteomics and phosphoproteomics to investigate the effects of a 60-day perinatal supplementation of Omega-3 fatty acids (ALA-rich flaxseed vs. saturated fat) on insulin sensitivity, immune function, and the endocannabinoid system (ECS) in dairy cows [50]. A proteomic analysis revealed that ALA supplementation reduced plasma inflammatory markers (IL-6, IL-17α) and inflammatory proteins in adipose tissue (TNFα, FAAH, MGLL, RELA, and AMPK) while improving systemic insulin sensitivity. Additionally, Omega-3 supplementation decreased ECS ligands and cannabinoid-1-receptor levels in peripheral blood mononuclear cells. Another study employed proteomics to examine the effects of high and low Omega-3 PUFA diets over 4 months on the hepatic proteomic profile of C57BL/6 mice [51]. High Omega-3 PUFA diets upregulated proteins involved in lipid metabolism (apolipoprotein A-I), carbohydrate metabolism (fructose-1,6-bisphosphatase, and ketohexokinase), and the citric acid cycle (TCA cycle, malate dehydrogenase, and GTP-specific succinyl CoA synthase). Meanwhile, it downregulated proteins such as regucalcin and adenosine kinase. These results indicate that Omega-3 PUFAs regulate multiple metabolic pathways, offering potential therapeutic targets for metabolic disorders.

High-caloric diets induce oxidative stress in the liver, which in turn causes protein carbonylation and protein damage. This series of events is closely associated with the development of metabolic diseases. A study in Sprague-Dawley (SD) rats fed a high-fat high-sucrose (HFHS) diet for 24 weeks used proteomics to explore the effects of EPA and DHA on hepatic protein carbonylation [52]. Omega-3 PUFA supplementation reduced protein carbonylation, enhanced antioxidant defenses, and modulated carbonylation levels of proteins involved in lipid metabolism (albumin), carbohydrate metabolism (pyruvate carboxylase), and oxidative stress response (catalase). These findings highlight the protective role of Omega-3 PUFAs in mitigating oxidative stress and metabolic damage induced by Westernized diets. Further research investigated the combined effects of marine Omega-3 PUFAs and grape seed polyphenols (GSE) in Wistar Kyoto rats fed an HFHS diet for 24 weeks [53]. A proteomic analysis revealed reduced protein carbonylation in both plasma and liver, with identified proteins showing modulation in lipid metabolism, detoxification, carbohydrate metabolism, and oxidative stress response. This study demonstrated a synergistic effect of Omega-3 PUFAs and polyphenols in alleviating oxidative protein damage caused by an obesogenic diet.

### 4.3. Animal Lipidomics: Effects of Omega-3 PUFAs on Lipid Profiles and Signaling Pathways

Over the past decade, research applied lipidomics explored tissue-specific effects, short-term and transgenerational supplementation, as well as the molecular mechanisms underlying the anti-inflammatory and metabolic benefits of Omega-3 PUFAs [54,55,56,57,58,59,60,61,62,63] (Appendix A).

Research shows that tissues exhibit distinct responses to fatty acid supplementation. A lipidomic study on mice supplemented with Omega-3 PUFAs analyzed plasma and nine tissues (liver, kidney, brain, white adipose, heart, lung, small intestine, skeletal muscle, and spleen), identifying 1026 lipid molecules [54]. Omega-3 PUFA intake significantly altered the lipid profiles in metabolic organs like the liver and kidney, but had minimal effects on the brain. Another study in C57BL/6 mice explored lipidomic changes in the liver, muscle, adipose tissue, and brain following supplementation with Omega-3 and conjugated linoleic acid (CLA)-enriched cheese [55]. This intervention reduced saturated fats, increased CLA and ALA levels (except in the brain), and modulated tissue-specific lipid and mitochondrial metabolism genes while lowering inflammatory gene expression. These findings highlight the tissue-specific effects of Omega-3 PUFAs and their potential to mitigate inflammation and prevent chronic diseases.

Different supplementation modes of Omega-3 PUFAs, such as short-term and transgenerational approaches, exert distinct effects on lipid metabolism. In a C57BL/6J mouse model, short-term (3 days) flaxseed oil (FS) supplementation increased Omega-3 PUFAs incorporation (e.g., C18:3α, EPA, and DHA) but failed to fully mitigate inflammation induced by a high-fat diet, highlighting its limited efficacy in suppressing inflammatory responses [56]. In contrast, a lipidomic analysis showed that transgenerational EPA supplementation significantly altered the skeletal muscle lipid composition. It increased the levels of phospholipids like PC 40:8 and PI 38:6 and decreased the levels of TG containing saturated fatty acids (such as palmitic acid and oleic acid) [57]. EPA also reduced the content of ceramides, alleviated lipotoxicity, and insulin resistance. These changes were closely associated with anti-inflammation, metabolic health, and the improvement of insulin sensitivity.

Lipidomics has been instrumental in uncovering the molecular mechanisms behind the anti-inflammatory effects of Omega-3 PUFA supplementation. In C57BL/6J mice fed high-fat diets, ALA-enriched butter (n3Bu) promoted the biosynthesis of long-chain Omega-3 PUFAs and their oxylipin metabolites, reducing hepatic triglyceride accumulation, adipose tissue inflammation, and improving insulin sensitivity [63]. These effects were linked to suppressed NF-κB activation and M1 macrophage polarization, emphasizing the importance of the Omega-6/Omega-3 PUFA ratio in modulating inflammation. Another study employed lipidomics to reveal that EPA is metabolized into 17,18-epoxyeicosatetraenoic acid (17,18-EpETE) and its derivative 12-hydroxy-17,18-epoxyeicosatetraenoic acid (12-OH-17,18-EpETE), which potently inhibit neutrophil infiltration and chemotaxis, demonstrating anti-inflammatory effects in a peritonitis model [64]. These studies illustrate how Omega-3 PUFAs exert anti-inflammatory effects through lipid remodeling and bioactive metabolite production.

### 4.4. Animal Metabolomics: Effects of Omega-3 PUFAs on Metabolic Pathways and Biomarkers

Compared with genomics, transcriptomics, and proteomics, metabolomics has the advantage of being closer to the phenotype, which can directly reflect the functional state of organisms. Metabolomics has been widely used to explore the effects of Omega-3 PUFAs on metabolic diseases [65,66,67,68,69,70] (Appendix A).

In a mouse model of early-stage NAFLD, a 4-day Omega-3 PUFA-enriched high-fat diet reversed hepatic lipid accumulation and increased the plasma levels of hydroxyeicosapentaenoic acids (HEPEs) and epoxyeicosatetraenoic acids (EEQs). These metabolites attenuated adipose tissue inflammation and suppressed pro-inflammatory cytokines and JNK pathway activation in macrophages, suggesting a potential preventative approach for NAFLD by targeting adipose tissue inflammation [71]. In another study, C57BL/6 mice with hyperhomocysteinemia (HHcy)-induced hepatic steatosis were supplemented with Omega-3 PUFAs for 6 weeks. Metabolomics revealed reduced hepatic ceramide levels and downregulated ceramide synthesis genes (Sptlc3, Degs2), indicating that Omega-3 PUFAs mitigate hepatic steatosis by modulating ceramide metabolism [65].

In SD rats, untargeted metabolomics showed that DHA-enriched phospholipids from large yellow croaker roe (LYCRPL) altered fecal metabolites, with 18 potential biomarkers including L-cysteine linked to lipid metabolism pathways such as the TCA cycle, glycolysis, and bile secretion. This suggests that LYCRPLs regulate lipid metabolism disorders by modulating these pathways [67]. Another study in mice fed high-fat diets with varying Omega-6/Omega-3 PUFA ratios for 12 weeks found that a high Omega-6/Omega-3 ratio improved body weight, insulin signaling, and mitochondrial function while reducing hepatic lipid accumulation. A metabolomic analysis revealed that a high Omega-3 PUFA intake upregulated the liver’s mitochondrial electron transport chain and TCA cycle pathways, enhancing mitochondrial complex activities, reducing fumaric acid levels, and alleviating oxidative stress [68].

The long-term effects of Omega-3 PUFAs on diabetic risk in the offspring of rats with gestational diabetes mellitus (GDM) were evaluated [69]. The GDM offspring fed Omega-3 PUFA-rich fish oil showed reduced oxidative stress, inflammation, and improved metabolic profiles compared to the Omega-3-deficient groups. Metabolomics revealed modulation of diabetes-related metabolites (e.g., ceramide, oxaloacetic acid, and ALA) and pathways, indicating that Omega-3 PUFAs mitigate diabetes risk in the GDM offspring by regulating metabolic and inflammatory processes.

In summary, metabolomics research has shown that Omega-3 PUFAs alleviate liver lipid accumulation, adipose tissue inflammation, and oxidative stress by regulating lipid metabolism (e.g., reducing ceramides and increasing HEPEs and EEQs) and improving mitochondrial function (e.g., enhancing the activity of the TCA cycle and the electron transport chain), thereby playing a metabolic protective role in NAFLD, hepatic steatosis, and the offspring of gestational diabetes.

### 4.5. Animal Microbiomics: Modulation of Gut Microbiota Composition and Function by Omega-3 PUFAs

Recent studies have shown that Omega-3 PUFAs significantly modulate gut microbiota and metabolic pathways [72,73,74,75,76,77,78,79,80,81]. Through multiple animal experiments, researchers have found that Omega-3 PUFAs can alleviate gut dysbiosis caused by high-fat, high-sugar western diets [72,73], maternal nutrition [74], polycystic ovary syndrome (PCOS) [75], and T2DM [77], reducing inflammation and oxidative stress, thereby improving metabolic health (Appendix A).

Researchers used microbiome analysis to study the effects of Omega-3 PUFAs from fish oil on the gut microbiome and metabolic pathways in C57BL/6 mice fed a beef-rich diet [72]. Omega-3 supplementation reduced serum triglyceride levels and altered gut microbial composition by decreasing potentially pathogenic bacteria (*Escherichia–Shigella*, *Mucispirillum*, *Helicobacter*, *Desulfovibrio*) while enhancing energy and glucose metabolism, as shown by the 16S rRNA analysis. In another study, an in vitro fecal fermentation model was used with cecal samples from rats fed a control diet (CD) or a high-fat high-sugar Western diet (WD) [73]. Treatment with a pomegranate oil mixture increased α-diversity and the relative abundances of *Firmicutes*, *Bacteroidetes*, *Akkermansia*, and *Blautia*, along with elevated levels of butyrate, acetate, tyrosine, and GABA. These changes indicate a positive modulation of the gut–brain axis and potential restoration of WD-induced microbiota imbalances.

Researchers also investigated the impact of maternal Omega-3 PUFA intake on the gut microbiota of female offspring in a BALB/c mouse model [74]. Using 16S rRNA gene sequencing, they found that maternal flaxseed oil consumption increased the proportions of *Rikenellaceae*, *Clostridium*, and *Oscillospira* in the offspring, with these changes persisting into adulthood. These alterations in gut microbiota were associated with enriched levels of metabolites such as 4-hydroxycinnamate sulfate, catechol sulfate, hippurate, and indolelactate in the visceral adipose tissue of adult female offspring, suggesting long-term metabolic programming effects of maternal Omega-3 intake.

Recently, Omega-3 PUFAs have been proven to benefit metabolic disorders in PCOS patients. In a dehydroepiandrosterone (DHEA)-PCOS mouse model, Omega-3 PUFAs increased the abundance of beneficial bacteria such as *Akkermansia* and *Alistipes*, alleviating gut dysbiosis and improving ovarian function and insulin resistance [75]. These microbial changes were associated with the reduction of ovarian inflammation and oxidative stress and the improvement of adipose tissue morphology and function, including the decrease of multilocular cells and thermogenic marker expressions like *Ucp1*, *Pgc1α*, *Cited*, and *Coc8b*.

Finally, in T2DM rat models, Omega-3 PUFAs from flaxseed oil reduced fasting blood glucose, inflammatory markers, and oxidative stress while modulating gut microbiota. Specifically, Omega-3 supplementation decreased *Firmicutes* and increased *Bacteroidetes* and *Alistipes* [77]. The relative abundance of *Firmicutes* was positively correlated with IL-1β, TNF-α, IL-6, IL-17a, and LPS, respectively. *Bacteroidetes* and *Alistipes* were negatively correlated with LPS. These findings suggest that Omega-3 PUFAs can alleviate T2DM by regulating gut microbiota and reducing inflammation.

## 5. Omics Approaches in Human Studies

### 5.1. Human Genomics and Epigenomics: Effects of Omega-3 PUFAs on Gene Expression and Epigenetics

Recent studies demonstrate that Omega-3 PUFAs can induce significant genomic and epigenomic changes, influencing metabolic pathways, inflammatory responses, and even transgenerational health outcomes [82,83,84] (Appendix A). These findings highlight the potential of Omega-3 PUFAs as a dietary intervention for preventing and managing metabolic diseases.

The Greenlandic Inuit have thrived for generations in harsh Arctic environments for centuries. They are confronted with frigid annual temperatures and rely on a specialized diet abundant in protein and fatty acids, especially Omega-3 PUFAs [82]. Researchers in a genomic study of the Greenlandic Inuit population identified significant adaptive signatures at the fatty acid desaturase (FADS) loci [82]. These genetic variations influence the levels of PUFAs and are associated with metabolic and anthropometric traits such as body weight and height. A membrane lipid analysis suggested that these alleles modulate fatty acid composition and may affect the regulation of growth hormone. This study highlights the genetic and physiological adaptations to Omega-3 PUFA-rich diets.

In an epigenomic study, overweight and obese subjects underwent Omega-3 PUFA supplementation for 6 weeks, leading to changes in DNA methylation at 308 CpG sites across 231 genes [83]. These methylation changes affected pathways related to inflammatory response, lipid metabolism, and cardiovascular signaling. Key genes such as AKT3, ATF1, HDAC4, and IGFBP5 showed methylation shifts correlated with improved plasma triglycerides, glucose, and cholesterol levels, suggesting that Omega-3 PUFAs modulate metabolic pathways critical for cardiovascular health.

Another epigenomic study examined DNA methylation in cord blood mononuclear cells from 118 mother–newborn pairs to explore the effects of maternal Omega-3 PUFA intake during pregnancy [84]. Differential methylation was observed at 8503 to 18,148 sites, depending on Omega-3 intake levels. Key pathways affected included signal transduction, metabolism, and immune response. Genes such as MSTN, IFNA13, ATP8B3, and GABBR2 showed methylation changes, potentially influencing insulin resistance, adiposity, and innate immune response in offspring. This study underscores the transgenerational epigenetic effects of Omega-3 PUFAs.

### 5.2. Human Transcriptomics: Effects of Omega-3 PUFAs on Gene Expression in Metabolic Health

This paper reviews several recent studies that have explored the transcriptomic changes in human subjects after exposure to Omega-3 PUFAs. These studies indicate that Omega-3 PUFAs trigger substantial transcriptomic changes in vascular endothelial cells, blood, and visceral adipocytes and regulate inflammatory, metabolic, and immune pathways [85,86,87,88] (Appendix A).

Researchers employed a DNA microarray analysis to study the impacts of Omega-3 PUFA, specifically DHA, on human umbilical vein endothelial cells (HUVECs) under pro-inflammatory conditions [85]. HUVECs were first treated with DHA and subsequently stimulated with interleukin IL-1β. DHA treatment regulated genes involved in immunological, inflammatory, and metabolic pathways, such as CYP4F2, TGF-β2, CD47, CARD11, and PDE5α. These findings reveal DHA’s role in modulating cardiovascular function, cellular growth, and inflammatory responses, highlighting its potential in managing metabolic and inflammatory diseases.

In a study involving obese women, a microarray transcriptome analysis of blood samples was conducted to assess the effects of 3-month Omega-3 PUFA supplementation [87]. Omega-3 PUFAs influenced PPAR-α, NRF2, and NF-κB target genes, significantly reducing inflammatory markers such as SELE, MCP-1, sVCAM-1, sPECAM-1, and hsCRP. These results suggest that Omega-3 PUFAs can mitigate inflammation and improve metabolic health in obese individuals.

Another study employed high-throughput sequencing to analyze the transcriptional responses of visceral adipocytes from healthy lean, obese, and colorectal cancer (CRC)-affected individuals exposed to Omega-3 and Omega-6 PUFAs [88]. Omega-3 PUFAs, particularly DHA, modulated specific long non-coding RNAs (lncRNAs) such as LIPE-AS1 and LUCAT1, which are implicated in obesity and metabolic diseases. These lncRNAs influenced pathways related to lipid metabolism, adipocyte function, and immune regulation, with impaired responses observed in obese and CRC groups compared to healthy individuals. The study highlights the potential of Omega-3 PUFAs to regulate metabolic health through lncRNA-mediated mechanisms.

### 5.3. Human Proteomics: Protein Biomarkers and Metabolic Effects of Omega-3 PUFAs

Omega-3 PUFAs have been extensively studied in various metabolic diseases, including metabolic syndrome, NAFLD, NASH (non-alcoholic steatohepatitis), and CVD using proteomic technologies, revealing their potential mechanisms of action [89,90,91,92,93,94,95] (Appendix A). In metabolic syndrome, a proteomic analysis was used to investigate the effects of Omega-3 PUFAs on subcutaneous white adipose tissue (WAT) in 75 MetS patients [89]. The LFHCC Omega-3 diet (a low-fat, high-complex carbohydrate diet supplemented with Omega-3 PUFAs) downregulated proteins associated with glucose metabolism, including annexin A2, gelsolin, and glycerol-3-phosphate dehydrogenase-1 (GPD1), suggesting improved insulin signaling and glucose metabolism. Another study employed proteomics to analyze the effects of different dietary lipid compositions on the proteome of peripheral blood mononuclear cells (PBMCs) in 24 MetS patients [90]. A two-dimensional proteomic analysis revealed that the LFHCC Omega-3 diet regulated proteins involved in immunological diseases and inflammatory responses, including nuclear FGB, FGG, VCL, and cytoplasmic ACTB, MACF1, and CAPZA1. These findings suggest that Omega-3 PUFAs may mitigate inflammation and oxidative stress, reducing cardiovascular disease risk in MetS patients.

Both NAFLD and NASH are frequently regarded as the hepatic presentations of metabolic syndrome. In NAFLD and NASH, proteomic studies have provided insights into the systemic effects of Omega-3 PUFAs. A study investigated the plasma proteome of 103 NAFLD patients treated with 3.36 g daily of DHA + EPA or placebo (olive oil) for 15–18 months [91]. A proteomic analysis indicated that DHA + EPA treatment affected pathways related to blood coagulation, immune/inflammatory responses, and cholesterol metabolism (*p* < 0.05), with reduced levels of prothrombin and apolipoprotein B-100, key proteins associated with cardiovascular risk. Another study used proteomics and lipidomics to explore the effects of Omega-3 PUFAs in 27 NASH patients over a 6-month treatment period [93]. Hepatic proteomic analyses revealed modifications in markers related to cell-matrix (FIBB, K1C9, PDIA6, TBA3E, and K2C75), lipid metabolism (PGRMC2 and FABPL), ER stress (HSPD1, EEF1A2, HNRPU, EEF2, RS27A, RL40, and UBB), and cellular respiratory pathways (PPIA, TPI1, ALDOB, GAPDH, PGM1, and ENO3). These findings highlight the potential of Omega-3 PUFAs in improving NASH through modulation of key metabolic pathways.

In CVD research, Omega-3 PUFAs have also demonstrated significant benefits. A study employing proteomics and systems biology investigated the effects of Omega-3-enriched milk in overweight healthy volunteers with a BMI of 25–35 kg/m^2^ [94]. Omega-3 PUFAs increased apolipoprotein E (Apo-E) in low-density lipoprotein (LDL). They enhanced several high-density lipoprotein (HDL)-associated proteins, including Apo A-I, LCAT, PON-1, Apo D, and Apo L1, improving lipid metabolic pathways. Another study used proteomics and lipidomics to analyze the effects of Omega-3 PUFAs on extracellular vesicles (EVs) in 40 patients with moderate CVD risks [95]. Omega-3 PUFAs reduced the number of circulating EVs by 27%, increased their Omega-3 PUFA content, and decreased their capacity to support thrombin generation by over 20%. Proteomic profiling of platelet-derived EVs indicated that Omega-3 PUFAs downregulated proteins involved in CVD pathogenesis, including proinflammatory and proatherosclerotic proteins such as RBP4, PF4V1, ESAM, and FBLN1, suggesting potential benefits in reducing thrombotic risk and improving CVD outcomes.

### 5.4. Human Metabolomics: Metabolic Impact and Therapeutic Potential of Omega-3 PUFAs in Various Health Conditions

Recent metabolomic studies have shown that Omega-3 PUFAs modulate lipid metabolism, reduce CVD, alleviate hepatic steatosis in NAFLD, and improve metabolic pathways in chronic inflammation and metabolic syndrome [96,97,98]. They also promote healthy aging by altering phospholipids and cholesterol esters and alleviate diabetic neuropathy pain by improving lipid metabolism [99,100] (Appendix A).

In 20 overweight and obese patients, researchers employed GC-MS and LC-MS to analyze plasma metabolites and investigate the effects of Omega-3 PUFAs (fish oil) and fenofibrate on lipid metabolism [96]. Both treatments reduced saturated TG, which is associated with a lower CVD risk. In addition, fish oil increases the levels of unsaturated TG, lysophosphatidylcholine (LPC), phosphatidylcholine, and cholesterol esters, which are further linked to a decreased CVD risk.

In NAFLD, a study utilized untargeted ultra-performance liquid chromatography-quadrupole/time-of-flight mass spectrometry (UPLC-Q-TOF-MSE) to investigate the serum metabolomic profiles of 96 patients following a 12-week intervention with a combination of phytosterol ester (PSE) and Omega-3 PUFAs [97]. Phytosterols are plant-derived sterols that can decrease LDL and improve metabolic disorders [101]. The combined supplementation significantly increased serum levels of phosphatidylcholine (PC) containing Omega-3 PUFAs, lysophosphatidylcholine (LysoPC), perillyl alcohol, and retinyl ester, which were negatively correlated with hepatic steatosis severity. This indicates that compared with either supplement alone, the combined use of PSE and Omega-3 PUFA supplements can more effectively improve metabolic disorders and alleviate hepatic steatosis.

The distinct effects of EPA and DHA were explored in a randomized, controlled, double-blind, crossover study involving 21 subjects with chronic inflammation and MetS [98]. A metabolomic analysis revealed that both EPA and DHA significantly affected the TCA cycle, glucuronate interconversion, and amino acid metabolism pathways. EPA specifically reduced the levels of fumarate and α-ketoglutarate, which are intermediates in the TCA cycle, while increasing the levels of glucuronate and non-esterified DHA. In contrast, DHA had a more pronounced impact on the TCA cycle, reducing the levels of multiple intermediates and increasing the levels of succinate and glucuronate.

There are also studies on the dynamic changes in the metabolomic profile of healthy old people after supplementing with Omega-3 PUFAs. A study utilized metabolomics and 1H-NMR spectroscopy to investigate the effects of Omega-3 PUFA on plasma metabolome in 12 young and 12 older healthy adults [100]. The results of plasma metabolomics suggest that there are subtle differences between healthy young and older adults. After the older adults were supplemented with Omega-3 PUFA, more significant changes occurred in phospholipids, cholesterol esters, diglycerides, and triglycerides.

Approximately 30–50% of diabetic patients will experience varying degrees of neuropathic pain [102]. In diabetic peripheral neuropathy, a study employed metabolomics to explore the effects of Omega-3 PUFA supplementation (1000 mg DHA and 200 mg EPA daily) over three months in 40 Mexican-American individuals with T2DM [99]. The intervention significantly improved neuropathic pain symptoms, as measured by the McGill Pain Questionnaire, with reduced plasma sphingosine levels and increased DHA correlating with improved pain scores. These findings suggest that Omega-3 PUFA supplementation may alleviate diabetic neuropathy pain in diabetes management.

### 5.5. Human Lipidomics: Modulation of Lipid Metabolism by Omega-3 PUFAs in Health and Disease

Omega-3 PUFAs have been extensively studied across various metabolic diseases using lipidomics, revealing their potential to modulate lipid profiles and improve human metabolic health [103,104,105,106,107,108,109] (Appendix A). In a longitudinal crossover clinical study involving 20 overweight participants, researchers investigated the effects of Omega-3 PUFAs on the HDL lipidome [103]. Participants consumed milk supplemented with Omega-3 PUFAs for 28 days. A lipidomic analysis showed that Omega-3 supplementation significantly increased DHA and EPA levels within HDL lipid species, particularly cholesteryl esters (CE), TG, and phosphatidylcholines, suggesting that dietary Omega-3 PUFAs can enrich HDL with beneficial fatty acids, potentially enhancing cardiovascular health.

In patients with MetS and NAFLD, a double-blind, placebo-controlled trial involving 60 participants examined the effects of 3.6 g/day Omega-3 PUFAs over one year [104]. A lipidomic analysis revealed that Omega-3 PUFA treatment significantly increased levels of Omega-3-enriched TGs and PLs, reduced serum GGT (Gamma-Glutamyl Transferase) activity, and decreased liver fat content, particularly in patients who also lost weight. These findings imply that long-term Omega-3 PUFA supplementation improves liver function and lipid metabolism in MetS and NAFLD.

A study on healthy individuals employed targeted lipidomics to analyze the effects of Omega-3 PUFAs on glycerophospholipids (GPs) and sphingolipids (SPs) over 21 days [105]. The results showed significant changes in lipid species, with lysophospholipids increasing after 3 days and phosphatidylserines exhibiting changes at later stages. Phosphatidylcholines and alkylphosphatidylcholines decreased on the 21st day. The study found that Omega-3 PUFAs can regulate the enzymes involved in the metabolism of lysophospholipids and phosphatidylserine, thereby influencing biomarkers such as creatine kinase MB isoenzyme (CK-MB), urea, and triglycerides.

In obese men with MetS, a study combined calorie restriction (CR) with Omega-3 PUFA-rich fish oil supplementation for 12 weeks [106]. Patients in the CR + fish oil group showed significant reductions in body weight, waist circumference, and TG levels, particularly. Elevated levels of specific lipid species, such as TG (60:9) containing docosapentaenoic acid, negatively correlated with MetS features like BMI and blood pressure, suggesting that Omega-3 PUFAs and CR can improve lipid metabolism and mitigate MetS.

A study investigated the effects of combined intervention with pioglitazone and EPA/DHA on lipid metabolism in overweight/obese type 2 diabetic patients already on metformin therapy [109]. Compared to single interventions, the combination improved serum EPA/DHA levels, insulin sensitivity, metabolic flexibility, and lipid metabolism during a meal test. While Omega-3 PUFAs alone modestly increased fasting glycemia and HbA1c, the combination prevented this effect and enhanced lipid metabolism, suggesting a synergistic strategy for managing metabolic dysfunction in T2DM.

### 5.6. Human Microbiomics: Modulation of the Gut Microbiome and Metabolic Outcomes by Omega-3 PUFAs

The following two studies delve into the specific changes in the gut microbiota in response to Omega-3 PUFA’s intake, providing valuable insights into their health benefits [110,111].

In a double-blind randomized controlled trial, researchers used metabolomics and a microbiome analysis to study the effects of plant-derived Omega-3 PUFAs on blood lipids and gut microbiota in 75 patients with marginal hyperlipidemia [110]. Over 3 months, Omega-3 PUFA supplementation significantly reduced total cholesterol (TC) levels and altered gut microbiota, increasing *Bacteroidetes* abundance and decreasing the *Firmicutes*-to-*Bacteroidetes* ratio. 

The study used 16S rRNA gene pyrosequencing to investigate the effects of different oil blends rich in MUFA and PUFA fatty acids on the gut microbiota of 25 volunteers at MetS risk [111]. While the oil treatments did not alter bacterial phyla composition, they influenced the gut microbiota at the genus level, particularly in obese participants. MUFA-rich diets increased the abundance of *Parabacteroides*, *Prevotella*, *Turicibacter*, and *Enterobacteriaceae*, while PUFA-rich diets favored *Isobaculum*. These changes suggest that MUFA and PUFA intake can modulate gut microbiota, potentially impacting metabolic health, especially in obese individuals

## 6. Mechanistic Insights into Omega-3 PUFAs and Omics

Omega-3 PUFAs are pivotal in modulating metabolic diseases through five key mechanisms: epigenetic regulation, oxidative stress reduction, gut microbiome modulation, mitochondrial function improvement, and inflammation control (Figure 5). Below is a detailed and logically structured summary of these mechanisms, supported by recent research findings.

### 6.1. Epigenetic Regulation

Epigenetics involves heritable changes in gene expression without altering the DNA sequence. Omega-3 PUFAs influence epigenetic mechanisms, such as DNA methylation and miRNA expression, to regulate metabolism-related genes. In terms of DNA methylation, Omega-3 PUFAs can alter specific genes’ methylation status, reducing pro-inflammatory gene methylation levels and increasing anti-inflammatory gene expression [83,84]. This not only alleviates inflammation, but also improves insulin sensitivity and lipid metabolism. For example, maternal Omega-3 PUFA intake during pregnancy affects offspring DNA methylation, influencing genes like MSTN, IFNA13, ATP8B3, and GABBR2, which are linked to insulin resistance, adiposity, and immune responses [84]. Regarding miRNA regulation, Omega-3 PUFAs regulate miRNA expression, which then targets and suppress the expression of metabolism-related genes post-transcriptionally. For instance, EPA inhibits miR-34a-5p, a negative regulator of Irs2, enhancing insulin sensitivity and reducing hepatic gluconeogenesis [48]. Additionally, different Omega-6/Omega-3 PUFA ratios result in the differential expression of miRNAs (e.g., ssc-miR-15b, ssc-miR-7142-3p), which target and regulate genes related to adipose and immunity (e.g., ELOVL6, RBP4, ARRDC3, and METTL21C), affecting pathways like lipolysis, obesity, and protein degradation [43]. Furthermore, miRNAs themselves are subject to epigenetic regulation, such as DNA methylation, RNA modifications, and histone modifications [112]. However, this review did not identify any studies investigating whether Omega-3 PUFAs modulate metabolic diseases via the mechanism of epigenetic modification of miRNAs. This could potentially represent a worthwhile avenue for future research.

### 6.2. Oxidative Stress Reduction

Oxidative stress, resulting from an imbalance between oxidation and antioxidant defenses, contributes to metabolic diseases. Omega-3 PUFAs mitigate oxidative stress through several mechanisms. Firstly, they inhibit telomere shortening by enhancing the activity of antioxidant enzymes like SOD and CAT, reducing oxidative damage and improving telomere length in metabolic disease models [69]. Secondly, Omega-3 PUFAs reduce protein carbonylation, a marker of oxidative damage [52,53]. For example, fish oil reduces the carbonylation of catalase, albumin, and Akr1d1, protecting against oxidative stress and improving cardiovascular health [52]. Finally, Omega-3 PUFAs modulate metabolites associated with oxidative stress, such as ceramide and hexadecenoic acid, further alleviating oxidative damage [65,69]. Omega-3 PUFAs alleviate oxidative stress by enhancing antioxidant enzyme activity (e.g., SOD and CAT) and reducing protein carbonylation, processes closely linked to lipid peroxidation and DNA damage (e.g., oxidative base modifications and strand breaks). Lipid peroxidation—a hallmark of ferroptosis, an iron-dependent cell death driven by lipid hydroperoxide accumulation—may be attenuated by Omega-3 PUFAs through the reduced susceptibility of polyunsaturated fatty acids to peroxidation, thereby mitigating ferroptotic cell death in metabolic tissues [113]. While existing studies highlight the role of Omega-3 PUFAs in ameliorating oxidative stress and chronic inflammation, their direct effects on DNA repair mechanisms and ferroptosis regulation remain underexplored. Future investigations into these pathways could uncover novel strategies to counteract both ferroptosis and genomic instability in metabolic diseases.

### 6.3. Modulation of Gut Microbiota and Metabolites

Omega-3 PUFAs significantly influence the gut microbiome and its metabolites, which contributes to metabolic health. They impact the microbial composition by promoting beneficial bacteria (e.g., *Bacteroidetes*, *Akkermansia*, and *Alistipes*) while inhibiting pathogens (e.g., *Escherichia–Shigella*, *Helicobacter*), improving gut barrier function and reducing inflammation [72,73,75]. Omega-3 PUFAs also enhance short-chain fatty acid (SCFA) production (e.g., butyrate, propionate), which has anti-inflammatory and metabolic benefits [77,114]. Additionally, they lower pro-inflammatory metabolites like LPS, further reducing systemic inflammation [77]. Moreover, Omega-3 PUFAs can modulate the gut–brain axis by influencing neurotransmitter production (e.g., GABA) and gut–brain communication, potentially improving mood and cognitive function [73].

### 6.4. Mitochondrial Function Improvement

Mitochondrial dysfunction is central to metabolic diseases, and Omega-3 PUFAs play a key role in restoring mitochondrial health. In terms of energy metabolism, Omega-3 PUFAs enhance mitochondrial respiratory chain activity, which leads to an increase in ATP production and an improvement in cellular energy supply [68]. Regarding mitochondrial biogenesis, Omega-3 PUFAs promote this biogenesis, increasing mitochondrial number and function. For example, a low Omega-6/Omega-3 ratio enhances fatty acid oxidation and beige adipogenesis via NR2F2 regulation [39]. Omega-3 PUFAs also mitigate oxidative stress by reducing ER stress and oxidative damage, which helps protect mitochondrial integrity [93]. Moreover, long-term Omega-3 PUFA supplementation exerts transgenerational effects by alleviating mitochondrial dysfunction across generations, as evidenced by reduced mitochondrial-related gene disturbances in offspring [41].

### 6.5. Inflammation Control

Chronic inflammation is a hallmark of metabolic diseases, and Omega-3 PUFAs exert potent anti-inflammatory effects. They can reduce pro-inflammatory cytokines (e.g., TNF-α, IL-6, and IL-1β) by inhibiting NF-κB and MAPK signaling pathways [58,71]. In addition, Omega-3 PUFAs modulate immune cells by promoting the polarization of macrophages toward the anti-inflammatory M2 phenotype, which helps reduce adipose tissue inflammation and insulin resistance [63]. Omega-3 PUFAs metabolites like 17,18-EEQ, 5-HEPE, and 9-HEPE also have anti-inflammatory effects as they suppress macrophage inflammation via JNK signaling, thereby ameliorating conditions like NAFLD [71]. Moreover, ALA supplementation has been shown to improve systemic insulin sensitivity in dairy cows compared to the controls, which demonstrates the beneficial effects of Omega-3 PUFAs on insulin function [50].

## 7. Challenges and Future Directions

In the omics research on the supplementation of Omega-3 PUFAs in the context of metabolic diseases, there are a series of research challenges, which are analyzed from the aspects of samples, technologies, data analysis, and the characteristics of diseases.

### 7.1. Interindividual and Disease-Stage Variability

The efficacy of Omega-3 PUFAs in metabolic diseases is influenced by interindividual variability and disease progression [115,116]. Genetic polymorphisms (e.g., FADS loci) and lifestyle factors (e.g., diet, smoking) significantly modulate Omega-3 absorption and metabolic responses, contributing to heterogeneous therapeutic outcomes [115,116,117,118,119]. Additionally, metabolic diseases exhibit dynamic pathophysiological changes across stages (e.g., NAFLD progression from steatosis to cirrhosis) and systemic interactions (e.g., diabetes affecting cardiovascular and neural systems), further complicating intervention studies [54,120,121,122]. Current omics technologies struggle to address these challenges due to limited capacity to capture real-time, system-wide interactions and representative sampling across diverse populations.

### 7.2. Challenges in Multi-Omics Technologies

Current omics technologies still face problems with detection sensitivity and accuracy [123,124]. In metabolomics, low-abundance metabolites important for the relationship between Omega-3 PUFAs and metabolic diseases may not be accurately detected [125], and proteomics has difficulties detecting and quantifying modified proteins [126]. Most existing omics technologies can only perform a static sample analysis, struggling to capture the dynamic mechanism of Omega-3 PUFAs in metabolic diseases [127]. A lack of unified technical standards and quality control systems across different research teams leads to poor data comparability and reproducibility for multi-omics joint analysis [128,129]. Multi-omics research generates high-dimensional data [130,131], and screening key information related to Omega-3 PUFAs and metabolic diseases while eliminating noise is a significant data analysis challenge.

### 7.3. Future Research Avenues: Direct Target Investigation

Most current omics research focuses on pathway analysis and the influence on prominent molecules’ expression levels or activities, while in-depth direct target studies are very limited. Discovering direct targets of Omega-3 PUFA supplementation in metabolic diseases is crucial for unveiling their beneficial mechanisms and developing precise therapies. Recent chemoproteomic technologies offer powerful tools for target identification, including Thermal Proteome Profiling (TPP) [132], Limited Proteolysis-Mass Spectrometry (LiP-MS) [133], and Activity-Based Protein Profiling (ABPP) [134,135]. These methods help uncover protein–ligand interactions and hold the potential for identifying Omega-3 PUFAs targets. Current research has identified several proteins as direct targets of fatty acids, including G Protein-Coupled Receptor 120 (GPR120) [136], Liver X Receptor (LXR) [137], and Peroxisome Proliferator-Activated Receptors (PPARs) α and γ [138]. Continuous exploration with advanced chemoproteomic techniques like TPP, LiP-MS, and ABPP is essential.

### 7.4. Future Research Avenues: Computational Biology Methods and Omega-3 Multi-Omics Integration

Constructing an Omega-3 PUFAs multi-omics database is very important. This database would centralize genomic, transcriptomic, proteomic, and metabolomic data, providing researchers with a one-stop data query and analysis platform. Additionally, developing computational models based on these omics data is highly significant. These models can not only accurately predict the therapeutic effects of Omega-3 PUFAs in various metabolic diseases such as obesity and hyperlipidemia, but also profoundly explore potential therapeutic targets, providing new directions for drug development. By simulating disease progression, these models can reveal key intervention points of Omega-3 PUFAs, aiding in early diagnosis and prevention. Furthermore, by evaluating the combined therapeutic effects of Omega-3 PUFAs with other treatments, treatment combinations can be optimized, improving efficacy and safety and bringing new opportunities for the research and treatment of metabolic diseases.

## 8. Conclusions

The global burden of metabolic diseases, including obesity, T2DM, and NAFLD, continues to rise, driven by lifestyle and dietary factors. Omega-3 PUFAs, particularly EPA and DHA, have emerged as key dietary components with significant potential to ameliorate these metabolic diseases. Advances in omics technologies, including transcriptomics, proteomics, lipidomics, and microbiomics, have provided deep insights into the molecular mechanisms by which Omega-3 PUFAs modulate epigenetics, inflammation, oxidative stress, mitochondrial function, and gut microbiota. Despite progress, challenges remain in understanding individual variability, dynamic metabolic processes, and the full spectrum of direct molecular targets.

Future research should prioritize advanced chemoproteomics (e.g., TPP, LiP-MS) and multi-omics integration to uncover novel targets and optimize personalized interventions. Individual variability in Omega-3 responses driven by genetic factors (e.g., FADS polymorphisms), gut microbiota dynamics, and epigenetic modifications necessitate tailored strategies. Integrating multi-omics data (genomics, lipidomics, and microbiomics) with clinical biomarkers could guide EPA/DHA dosing, microbiome-directed co-interventions (e.g., prebiotics), and combinatorial therapies (e.g., Omega-3 with calorie restriction). Computational modeling of metabolic networks and precision phenotyping will enable stratified interventions, advancing precision nutrition paradigms for metabolic disease management.

## Figures and Tables

**Figure 1 nutrients-17-01836-f001:**
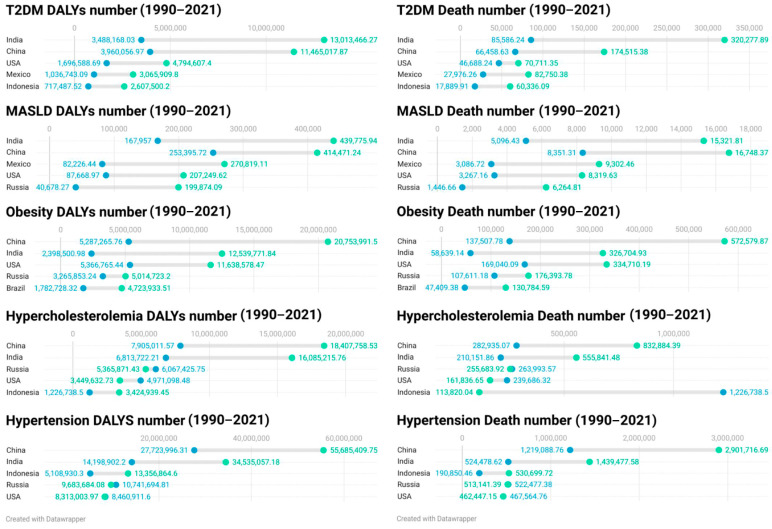
Trends in the five countries with the highest absolute burden of five metabolic diseases, 1990–2021 (Based on DALYs and mortality data).

**Figure 2 nutrients-17-01836-f002:**
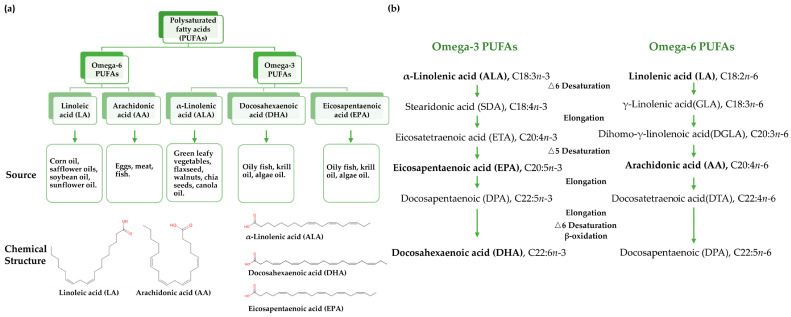
Polyunsaturated fatty acids and dietary sources. (**a**) Chemical structure and dietary sources; (**b**) general metabolic pathway for Omega-3 and Omega-6 PUFAs.

**Figure 3 nutrients-17-01836-f003:**
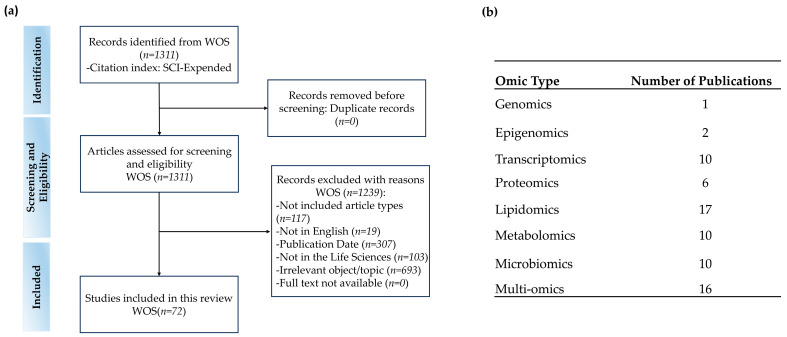
PRISMA flowchart of study inclusion and exclusion and distribution of publication numbers by omic types for included studies. (**a**) Eligibility criteria for the selection of research articles. (**b**) Distribution of publication numbers by omic types for the 72 included studies.

**Figure 4 nutrients-17-01836-f004:**
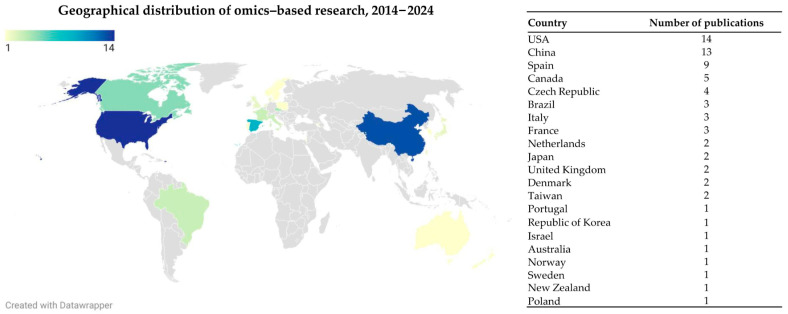
Geographical distribution of omics-based research publications on Omega-3 PUFAs in animal or human models (ranked by publication volume by country).

**Figure 5 nutrients-17-01836-f005:**
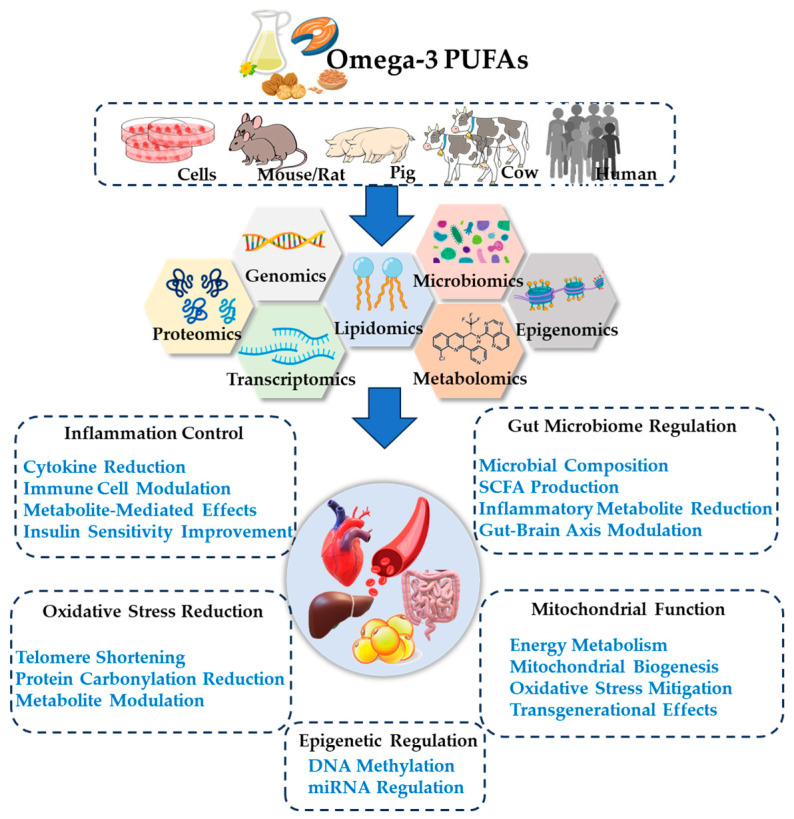
Mechanistic insights into Omega-3 PUFAs in metabolic diseases through omics analysis in animal and human models.

**Table 1 nutrients-17-01836-t001:** Exclusion criteria of selected articles.

No.	Exclusion Criteria
1	Excluded article types, e.g., review, proceedings, feature, editorial material
2	Not written in English
3	Publication date outside 2014–2024
4	Not in the field of life sciences
5	Irrelevant objects or topics
6	Without available full-text

## Data Availability

All data are included in the manuscript and its Appendix A.

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
