# Peer review of "Dietary Omega-3 PUFAs in Metabolic Disease Research: A Decade of Omics-Enabled Insights (2014–2024)"

_nutrients, 2025, doi:10.3390/nu17111836_

Round 1
Reviewer 1 Report
Comments and Suggestions for Authors
The authors of the article entitled Dietary Omega-3 PUFAs in Metabolic Disease Research: A Decade of Omics-Enabled Insights (2014-2024) took into consideration an important healthcare problem – Nutrition.
The increased speed of life forces the high-processed/converted food rich in saturated fat acids and sugars, which leads to overweight and, finally, obesity. The last one is the parent of whole civilisation diseases such as neurodegenerative, cardiovascular, diabetes and cancers. The main point is the lack of the correct amount of unsaturated fat acids, Omega-3, which, in ratio to Omega-6, should be 1 to 4, but the observed amount is 1 to 20.
Therefore, from this point of view, the article is valuable and interesting.
However, I have some critical remarks:
The article is not so well written – I recommend that authors shorten the text and simplify their idea
The resolution of the figures should be increased, and the size should be extended in the current form their mining is unreadable.
All tables should be shifted to the supplementary materials
The influence of unsaturated acids quality in the context of ferroptosis and DNA damage should be discussed.
The references are correctly selected and cited and don’t cause confusion.
In conclusion, I cannot recommend the article for publication in its present form. I believe that the authors will make an effort to answer my question.
Author Response
Dear editor and reviewers,
Thank you very much for your valuable comments and suggestions on our manuscript entitled “Dietary Omega-3 PUFAs in Metabolic Disease Research: A Decade of Omics-Enabled Insights (2014-2024)” to the special issue “Special Issue: Dietary Fatty Acids and Metabolic Health” on Nutrients. We have carefully considered each of your comments and made the necessary revisions to improve the quality of our manuscript. Please find our detailed responses below:
The authors of the article entitled Dietary Omega-3 PUFAs in Metabolic Disease Research: A Decade of Omics-Enabled Insights (2014-2024) took into consideration an important healthcare problem-Nutrition. The increased speed of life forces the high-processed/converted food rich in saturated fat acids and sugars, which leads to overweight and, finally, obesity. The last one is the parent of whole civilisation diseases such as neurodegenerative, cardiovascular, diabetes and cancers. The main point is the lack of the correct amount of unsaturated fat acids, Omega-3, which, in ratio to Omega-6, should be 1 to 4, but the observed amount is 1 to 20. Therefore, from this point of view, the article is valuable and interesting. However, I have some critical remarks:
Comment 1. The article is not so well written – I recommend that authors shorten the text and simplify their idea.
Response 1:
We sincerely appreciate this constructive suggestion. The manuscript has now been thoroughly revised to improve clarity and conciseness. Specifically, we have removed some content from the Introduction and Discussion sections, and moved Tables 2 and 3 to the supplementary materials, significantly reducing the word count of the manuscript. We believe these changes have significantly enhanced the overall quality of the paper.
Comment 2. The resolution of the figures should be increased, and the size should be extended in the current form their mining is unreadable.
Response 2:
Thank you for highlighting this issue. All figures have been re-generated in high resolution and resized to ensure optimal clarity. Additionally, labels, axes, and annotations have been adjusted for improved legibility. The revised figures are as follows:
Figure 1
Figure 2
Figure 3
Figure 4
Figure 5
Comment 3. All tables should be shifted to the supplementary materials
Response 3:
As suggested, Table 2 and Table 3 have been moved to the Supplementary Materials section (now labeled as Supplementary Tables S1and Table S2). This adjustment helps focus the narrative while ensuring all data remains accessible.
Comment 4. The influence of unsaturated acids quality in the context of ferroptosis and DNA damage should be discussed.
Response 4:
In Section 6.2, we discussed the critical role of unsaturated fatty acids (UFAs) in ferroptosis and DNA damage, with the main content including:
“Omega-3 PUFAs alleviate oxidative stress by enhancing antioxidant enzyme activity (e.g., SOD and CAT) and reducing protein carbonylation, processes closely linked to lipid peroxidation and DNA damage (e.g., oxidative base modifications and strand breaks). Lipid peroxidation-a hallmark of ferroptosis, an iron-dependent cell death driven by lipid hydroperoxide accumulation-may be attenuated by Omega-3 PUFAs through reduced susceptibility of polyunsaturated fatty acids to peroxidation, thereby mitigating ferroptotic cell death in metabolic tissues. While existing studies highlight the role of Omega-3 PUFAs in ameliorating oxidative stress and chronic inflammation, their direct effects on DNA repair mechanisms and ferroptosis regulation remain underexplored. Future investigations into these pathways could uncover novel strategies to counteract both ferroptosis and genomic instability in metabolic diseases.”
Comment 5. The references are correctly selected and cited and don’t cause confusion.
Response 5:
We thank the reviewer for this positive remark. All references have been carefully verified to ensure accuracy, relevance, and balanced representation of the field. In-text citations and the reference list remain consistent with journal guidelines.
In conclusion, I cannot recommend the article for publication in its present form. I believe that the authors will make an effort to answer my question.
Response:
We sincerely appreciate the time and effort the reviewer has dedicated to evaluating our manuscript and providing constructive feedback. We have carefully addressed all raised concerns in the revised version. We sincerely hope the reviewer will reconsider their recommendation based on these comprehensive revisions.
Thank you again for your valuable feedback. We believe that these revisions have significantly improved the manuscript and addressed all the concerns raised. We hope that you will find our revised manuscript suitable for publication.
Sincerely,
Professor Hsien-Da Huang, Ph.D. (Corresponding author)
The Chinese University of Hong Kong, Shenzhen, H.L. Tu Building, 2001 Longxiang Blvd., Longgang District, Guangdong 518172, China.
E-mail: huanghsienda@cuhk.edu.cn
Homepage: https://warshel.cuhk.edu.cn/~hdhuang/pi.php?lang=en

Reviewer 2 Report
Comments and Suggestions for Authors
Dear Authors;
Re: [Manuscript ID: nutrients-3593548]
Title: Dietary Omega-3 PUFAs in Metabolic Disease Research: A Decade of Omics-Enabled Insights (2014-2024)
Please find my comments / suggestions below:
1. In this manuscript you aimed to review (as stated in the ABSTRACT) the importance of Omega-3 PUFAs (based on omics technologies) in metabolic health and outline future directions for research and clinical applications.
The manuscript is very well written. However, most of the Figures are not adequately legible or are of low resolution.
2. On top of the manuscript, article type is mentioned as "Review". However, it is not clear that if it is a meta analysis, narrative review, or a systematic review (there is a section on MATERIALS AND METHODS in the paper).
3. The concept of "personalised intervention strategies" is somehow overlooked, while mentioned in the Abstract. Please make these clear for the readers.
4. The time frame of the study is between 2014 to 2024. However, there are big number of References which are outside these limits.
5. Figure 1 legend has an internet link when we put mouse cursor on it. What is the reason for this "link"? What is the situation of Copyright of this Figure? Please enhance its resolution.
6. The chemical structures in Figure 2 are too small. However, it is a very useful illustration relevant to your study and the context of your manuscript.
7. Table 1 legend says "Eligibility criteria". However, it seems that column 2 lists "excluding criteria" and only the 1st raw mentions these. Please clarify.
8. Sections (a) and (b) are not explained in the Figure 3 legend .
9. Some paragraphs left with no References (if other papers' methodologies are used need to cite them, otherwise please make it clear it is your own approach / method). For instance, check paragraphs in sections 2.3. and 3. as well as paragraphs ending in Lines 197 and 206. Please also double-check the last 2 References dated 1997 and 2001.
10. In Figure 4 what is the date interval of the reported data? Even if it is between the dates mentioned in the Title, please make it clear on the Figure.
11. Figure 5 is outstanding.
12.The List of Abbreviations is extremely useful. Can you list the items in an alphabetical order?
Thank you and all the best in your future studies.
Author Response
Dear editor and reviewer,
Thank you very much for your valuable comments and suggestions on our manuscript entitled “Dietary Omega-3 PUFAs in Metabolic Disease Research: A Decade of Omics-Enabled Insights (2014-2024)” to the special issue “Special Issue: Dietary Fatty Acids and Metabolic Health” on Nutrients. We have carefully considered each of your comments and made the necessary revisions to improve the quality of our manuscript. Please find our detailed responses below:
Comment 1. In this manuscript you aimed to review (as stated in the ABSTRACT) the importance of Omega-3 PUFAs (based on omics technologies) in metabolic health and outline future directions for research and clinical applications. The manuscript is very well written. However, most of the Figures are not adequately legible or are of low resolution.
Response 1:
We have regenerated all figures in high resolution and adjusted their size, fonts, and labels for optimal clarity. Each figure has been carefully reviewed to ensure readability.
Comment 2. On top of the manuscript, article type is mentioned as "Review". However, it is not clear that if it is a meta analysis, narrative review, or a systematic review (there is a section on MATERIALS AND METHODS in the paper).
Response 2:
We have clarified in the Introduction and Materials and Methods sections that this is a systematic review with narrative synthesis. The Materials and Methods section describes the search strategy, inclusion/exclusion criteria, and data extraction process to align with PRISMA guidelines where applicable.
Comment 3. The concept of "personalised intervention strategies" is somehow overlooked, while mentioned in the Abstract. Please make these clear for the readers.
Response 3:
We appreciate the reviewer's valuable suggestion regarding the clarification of personalized intervention strategies. In response, we have substantially expanded the discussion of this concept in the concluding section (Section 8) of our review. The added content is as follows:
“Future research should prioritize advanced chemoproteomics (e.g., TPP, LiP-MS) and multi-omics integration to uncover novel targets and optimize personalized interventions. Individual variability in Omega-3 responses—driven by genetic factors (e.g., FADS poly-morphisms), gut microbiota dynamics, and epigenetic modifications—necessitates tai-lored strategies. Integrating multi-omics data (genomics, lipidomics, microbiomics) with clinical biomarkers could guide EPA/DHA dosing, microbiome-directed co-interventions (e.g., prebiotics), and combinatorial therapies (e.g., Omega-3 with calorie restriction). Computational modeling of metabolic networks and precision phenotyping will enable stratified interventions, advancing precision nutrition paradigms for metabolic disease management.”
Comment 4. The time frame of the study is between 2014 to 2024. However, there are big number of References which are outside these limits.
Response 4:
We appreciate the reviewer's observation regarding reference timeframes. Our study focuses on summarizing omics-based discoveries concerning dietary omega-3 PUFAs in metabolic diseases from 2014-2024, which is reflected in the primary results sections (Sections 4-6).
For other manuscript sections (Introduction, Methods, and Future Directions), while we preferentially cited recent publications (<5 years), we retained several older but seminal references (original Refs 118, 121, 124, 129, 130, 134-136) that either:
1) Represent foundational methodological approaches
2) Contain landmark theoretical frameworks
3) Report pivotal discoveries that remain current
These classic references constitute only 5.8% (8/136) of our total citations. However, in response to your comment, we have replaced several older references with more contemporary studies where possible without compromising scientific accuracy. We believe this balanced approach maintains historical context while emphasizing current research.
Comment 5. Figure 1 legend has an internet link when we put mouse cursor on it. What is the reason for this "link"? What is the situation of Copyright of this Figure? Please enhance its resolution.
Response 5:
Thank you for your suggestion. However, I can't find the link you mentioned in the submitted manuscript. Figure 1 is a graph I created using Datawrapper with data from the GBD database. Users have full copyright of the chart, but the free version requires retaining the Datawrapper attribution. Moreover, the watermark on Figure 1 indicates that it was created with Datawrapper. The resolution of Figure 1 has been improved.
Figure 1
Comment 6. The chemical structures in Figure 2 are too small. However, it is a very useful illustration relevant to your study and the context of your manuscript.
Response 6:
We have enlarged and improved the chemical structures in Figure 2.
Figure 2
Comment 7. Table 1 legend says "Eligibility criteria". However, it seems that column 2 lists "excluding criteria" and only the 1st raw mentions these. Please clarify.
Response 7:
We have revised "Eligibility criteria" in Table 1 to “Exclusion Criteria”.
Comment 8. Sections (a) and (b) are not explained in the Figure 3 legend.
Response 8:
The legend now includes a detailed description of panels (a) and (b), linking them to the respective results in Section 3.
Figure 3. PRISMA flowchart of study inclusion and exclusion and distribution of publication numbers by omic types for included studies. (a) Eligibility criteria for the selection of research articles. (b) Distribution of publication numbers by omic types for the 72 included studies.
Comment 9. Some paragraphs left with no References (if other papers' methodologies are used need to cite them, otherwise please make it clear it is your own approach / method). For instance, check paragraphs in sections 2.3. and 3. as well as paragraphs ending in Lines 197 and 206. Please also double-check the last 2 References dated 1997 and 2001.
Response 9:
We sincerely appreciate the reviewer's careful scrutiny of our references. In response to your concerns:
We have thoroughly reviewed sections 2.3 and 3, and added all necessary references to support methodological descriptions and key claims.
We have supplemented the appropriate citations to relevant literature for the paragraphs ending at lines 197 and 206, as well as for the entire manuscript.
Regarding the 1997 and 2001 references: These seminal papers were intentionally retained as they represent foundational discoveries in the field that established key molecular targets. While newer studies exist, these original works remain essential citations when discussing the historical development of these research areas.
Comment 10. In Figure 4 what is the date interval of the reported data? Even if it is between the dates mentioned in the Title, please make it clear on the Figure.
Response 10:
Added a date range label (2014–2024) directly to the figure.
Figure 4. Geographical distribution of omics-based research publications on Omega-3 PUFAs in animal or human models (Ranked by publication volume by country)
Comment 11. Figure 5 is outstanding.
Response 11:
We thank the reviewer for this positive feedback.
Comment 12. The List of Abbreviations is extremely useful. Can you list the items in an alphabetical order? Thank you and all the best in your future studies.
Response 12:
The list has been reorganized alphabetically and cross-checked for consistency with the main text.
Thank you again for your valuable feedback. We believe that these revisions have significantly improved the manuscript and addressed all the concerns raised. We hope that you will find our revised manuscript suitable for publication.
Sincerely,
Professor Hsien-Da Huang, Ph.D. (Corresponding author)
The Chinese University of Hong Kong, Shenzhen, H.L. Tu Building, 2001 Longxiang Blvd., Longgang District, Guangdong 518172, China.
E-mail: huanghsienda@cuhk.edu.cn
Homepage: https://warshel.cuhk.edu.cn/~hdhuang/pi.php?lang=en

Reviewer 3 Report
Comments and Suggestions for Authors
This review manuscript is an interesting and well-elaborated compilation of the current knowledge of the medicinal properties of PUFAs. The content is meaningful and the work covers the important and relevant findings in the field. Some stylistic issues need to be cared for in a revision of this manuscript before it can be accepted for publication:
Figure 1: Please replace ´´Mexican´´ by ´´Mexico´´.
Figure 2a: The chemical structures of the PUFAs need to be modified and corrected.
Figure 3: Please correct ´´Records identified through from WOS´´.
Table 2: The animal column requires some modification (Iberian×Du … roc, Dawley(SD … )).
Table 3: The ´´Durati … on´´ column needs some modification (month … s).
References: Please adjust the references according to the journal style.
Author Response
Dear editor and reviewers,
Thank you very much for your valuable comments and suggestions on our manuscript entitled “Dietary Omega-3 PUFAs in Metabolic Disease Research: A Decade of Omics-Enabled Insights (2014-2024)” to the special issue “Special Issue: Dietary Fatty Acids and Metabolic Health” on Nutrients. We have carefully considered each of your comments and made the necessary revisions to improve the quality of our manuscript. Please find our detailed responses below:
This review manuscript is an interesting and well-elaborated compilation of the current knowledge of the medicinal properties of PUFAs. The content is meaningful and the work covers the important and relevant findings in the field. Some stylistic issues need to be cared for in a revision of this manuscript before it can be accepted for publication:
Comment 1. Figure 1: Please replace ´´Mexican´´ by ´´Mexico´´.
Response 1:
We have corrected the term from "Mexican" to "Mexico" in Figure 1 to ensure geographical accuracy.
Figure 1
Comment 2. Figure 2a: The chemical structures of the PUFAs need to be modified and corrected.
Response 2:
The chemical structures in Figure 2a have been carefully revised and verified using ChemSpider, the Royal Society of Chemistry's authoritative database for validated chemical structures. The updated figure now provides clear, high-resolution representations of the PUFAs.
Figure 2
Comment 3. Figure 3: Please correct ´´Records identified through from WOS´´.
Response 3:
The text in Figure 3 has been revised to: "Records identified from Web of Science (WOS)".
Figure 3
Comment 4. Table 2: The animal column requires some modification (Iberian×Du … roc, Dawley(SD … )).
Response 4:
The "Animal" column in the original Table 2 has been reformatted as requested.
(Note: Following other reviewers' suggestions to improve readability, Table 2 has been moved to Supplementary Materials as Table S1 while maintaining full data accessibility.)
Comment 5. Table 3: The ´´Durati … on´´ column needs some modification (month … s).
Response 5:
The "Duration" column in Table 3 has been reformatted as requested.
(Note: Following other reviewers' suggestions to improve readability, Table 3 has been moved to Supplementary Materials as Table S2 while maintaining full data accessibility.)
Comment 6. References: Please adjust the references according to the journal style.
Response 6:
All references have been reformatted to comply with the journal’s citation style.
Thank you again for your valuable feedback. We believe that these revisions have significantly improved the manuscript and addressed all the concerns raised. We hope that you will find our revised manuscript suitable for publication.
Sincerely,
Professor Hsien-Da Huang, Ph.D. (Corresponding author)
The Chinese University of Hong Kong, Shenzhen, H.L. Tu Building, 2001 Longxiang Blvd., Longgang District, Guangdong 518172, China.
E-mail: huanghsienda@cuhk.edu.cn
Homepage: https://warshel.cuhk.edu.cn/~hdhuang/pi.php?lang=en

Reviewer 4 Report
Comments and Suggestions for Authors
This is ancomprehensive review article of high quality and novelty. It focuses on the omics technologies and their applications for evaluating the impact of omega-3 PUFAs in metabolic diseases. Even if its length is too high and quite tiring for the readers, it is well-written and well-organized. Some points should be addressed.
- The authors should try to add more information in the Abstract by providing distinctly the Background/Objectives, Methods, Results and Conclusions of their study.
- The resolution of Figure 1 should be improved.
- The resolution of Figure 2 should also be improved. It could also be split into two figures with higher size to be more easily readable.
- The sentence in lines 193-197 "Over the past decade, the application of transcriptomics ... overall impact on metabolic health holistically." is too long and it should be split into two smaller sentences in order to be more easily readable.
- At the end of the section 4.1, a relevant reference should be added. In this paragraph, a bit more information should be added concerning the omega-3/omega-6 ratio and reporting why this ratio is very important.
- Please, avoid words such as "discovered" (lines 232-233). It could be used the statement "They found...".
- The sentence in lines 253-258 is to long and it should be split into two smaller sentences.
- Please, provide a bit more details for Inuit population.
- Concerning the study cited as [101], lines 562-5571, the role of phytosterols should brifly described.
Author Response
Dear editor and reviewers,
Thank you very much for your valuable comments and suggestions on our manuscript entitled “Dietary Omega-3 PUFAs in Metabolic Disease Research: A Decade of Omics-Enabled Insights (2014-2024)” to the special issue “Special Issue: Dietary Fatty Acids and Metabolic Health” on Nutrients. We have carefully considered each of your comments and made the necessary revisions to improve the quality of our manuscript. Please find our detailed responses below:
This is a comprehensive review article of high quality and novelty. It focuses on the omics technologies and their applications for evaluating the impact of omega-3 PUFAs in metabolic diseases. Even if its length is too high and quite tiring for the readers, it is well-written and well-organized. Some points should be addressed.
Comment 1. The authors should try to add more information in the Abstract by providing distinctly the Background/Objectives, Methods, Results and Conclusions of their study.
Response 1:
We have restructured the Abstract as follows:
“Background/Objectives: The rising global prevalence of metabolic diseases (e.g., obesity, type 2 diabetes mellitus) underscores the need for effective interventions. Omega-3 polyunsaturated fatty acids (PUFAs) exhibit therapeutic potential, yet their molecular mechanisms remain unclear. This systematic review synthesizes a decade (2014–2024) of omics research to elucidate Omega-3 PUFA mechanisms in metabolic diseases and identify future directions.
Methods: A PRISMA-guided search of Web of Science identified studies on Omega-3 PUFAs, metabolic diseases, and omics. After excluding reviews, non-English articles, and irrelevant studies, 72 articles were analyzed (16 multi-omics, 17 lipidomics, 10 transcriptomics/metabolomics/microbiomics each, 6 proteomics).
Results: Omics studies demonstrated that Omega-3 PUFAs, particularly EPA and DHA, improve metabolic health through interconnected mechanisms. They regulate epigenetic processes, including DNA methylation and miRNA expression, influencing genes linked to inflammation and insulin sensitivity. Omega-3 PUFAs reduce oxidative stress by mitigating protein carbonylation and enhancing antioxidant defenses. Gut microbiota modulation is evident through increased beneficial taxa (e.g., Bacteroidetes, Akkermansia) and reduced pro-inflammatory species, correlating with improved metabolic parameters. Mitochondrial function is enhanced via upregulated fatty acid oxidation and TCA cycle activity, while anti-inflammatory effects arise from NF-κB pathway suppression and macrophage polarization toward an M2 phenotype. Challenges include interindividual variability in responses and limited understanding of dynamic metabolic interactions.
Conclusions: Omega-3 PUFAs target multiple pathways to improve metabolic health. Future research should prioritize chemoproteomics for direct target identification, multi-omics integration, and personalized strategies combining Omega-3 with therapies like calorie restriction.”
Comment 2. The resolution of Figure 1 should be improved.
Response 2:
Thank you for highlighting this issue. Figure 1 has been re-generated in high resolution and resized to ensure optimal clarity. The revised versions are as follows:
Figure 1
Comment 3. The resolution of Figure 2 should also be improved. It could also be split into two figures with higher size to be more easily readable.
Response 3:
Thank you for highlighting this issue. Figure 2 has been re-generated in high resolution and resized to ensure optimal clarity. The revised versions are as follows:
Figure 2
Comment 4. The sentence in lines 193-197 "Over the past decade, the application of transcriptomics ... overall impact on metabolic health holistically." is too long and it should be split into two smaller sentences in order to be more easily readable.
Response 4:
The original sentence has been split into two clearer statements:
Over the past decade, the application of transcriptomics in the study of Omega-3 PUFAs has gained momentum. Key areas include the transgenerational effects of Omega-3 PUFAs, miRNA-mediated regulatory mechanisms, and the integration of transcriptomics with other omics technologies to understand their overall impact on metabolic health holistically (Table S1).
Comment 5. At the end of the section 4.1, a relevant reference should be added. In this paragraph, a bit more information should be added concerning the omega-3/omega-6 ratio and reporting why this ratio is very important.
Response 5:
We have added reference and supporting information for omega-3/omega-6 ratio importance:
“Excessive Omega-6 PUFAs in commercial pig diets raise inflammatory disease risks. Balancing Omega-6/Omega-3 PUFAs ratios regulates Omega-6 metabolites as Omega-3 counteracts Omega-6's pro-inflammatory response.”
Comment 6. Please, avoid words such as "discovered" (lines 232-233). It could be used the statement "They found...".
Response 6:
We have replaced "discovered" with "found" and performed a full manuscript scan to ensure consistent, appropriate terminology throughout.
Comment 7. The sentence in lines 253-258 is too long and it should be split into two smaller sentences.
Response 7:
The original sentence has been split into two clearer statements:
“High Omega-3 PUFA diets upregulated proteins involved in lipid metabolism (apolipoprotein A-I), carbohydrate metabolism (fructose-1,6-bisphosphatase, ketohexokinase), and the citric acid cycle (TCA cycle, malate dehydrogenase, GTP-specific succinyl CoA synthase). Meanwhile, it downregulated proteins such as regucalcin and adenosine kinase.”
Comment 8. Please, provide a bit more details for Inuit population.
Response 8:
We have added details for Greenlandic Inuit population:
“The Greenlandic Inuit have thrived for generations in harsh Arctic environment for centuries. They are confronted with frigid annual temperatures and rely on a specialized diet abundant in protein and fatty acids, especially Omega-3 PUFAs.”
Comment 9. Concerning the study cited as [101], lines 562-571, the role of phytosterols should briefly described.
Response 9:
We have added phytosterols information in line 580-581:
“Phytosterols are plant-derived sterols that can decrease LDL and improve metabolic dis-orders.”
Thank you again for your valuable feedback. We believe that these revisions have significantly improved the manuscript and addressed all the concerns raised. We hope that you will find our revised manuscript suitable for publication.
Sincerely,
Professor Hsien-Da Huang, Ph.D. (Corresponding author)
The Chinese University of Hong Kong, Shenzhen, H.L. Tu Building, 2001 Longxiang Blvd., Longgang District, Guangdong 518172, China.
E-mail: huanghsienda@cuhk.edu.cn
Homepage: https://warshel.cuhk.edu.cn/~hdhuang/pi.php?lang=en

Round 2
Reviewer 1 Report
Comments and Suggestions for Authors
The article, after correction, is suitable for publication, in my opinion.
Author Response
Comment: The article, after correction, is suitable for publication, in my opinion.
Response: Thanks for your positive comment!